# Synthesis of Bio-Based Thermoset Mixture Composed of Methacrylated Rapeseed Oil and Methacrylated Methyl Lactate: One-Pot Synthesis Using Formed Methacrylic Acid as a Continual Reactant

**DOI:** 10.3390/polym15081811

**Published:** 2023-04-07

**Authors:** Vojtěch Jašek, Jan Fučík, Veronika Melcova, Silvestr Figalla, Ludmila Mravcova, Štěpán Krobot, Radek Přikryl

**Affiliations:** 1Institute of Materials Chemistry, Faculty of Chemistry, Brno University of Technology, 61200 Brno, Czech Republic; 2Institute of Environmental Chemistry, Faculty of Chemistry, Brno University of Technology, 61200 Brno, Czech Republic

**Keywords:** methacrylated methyl lactate, viscosity modification, thermoset mixture, polymerizable precursors, resin, modified lactate

## Abstract

Methacrylated vegetable oils are promising bio-based polymerizable precursors for potential material application in several fields, such as coating technologies or 3D printing. The reactants’ availability for their production is an enormous advantage, but the modified oils also exhibit high apparent viscosity values and poor mechanical properties. This work focuses on a way to produce oil-based polymerizable material precursors in a mixture with a viscosity modifier in a one-batch process. The required methacrylic acid for the modification of epoxidized vegetable oils can be obtained as a secondary product of the methacrylation of methyl lactate forming a polymerizable monomer along with the acid. This reaction results in a yield of over 98% of methacrylic acid. Epoxidized vegetable oil can be added into the same batch using acid for oil modification which results in the one-pot mixture of both methacrylated oil and methyl lactate. The structural verifications of products were provided via FT-IR, ^1^H NMR, and volumetric methods. This two-step reaction process produces a thermoset mixture with a lower apparent viscosity of 142.6 mPa·s in comparison with methacrylated oil exhibiting a value of 1790.2 mPa·s. Other physical-chemical properties of the resin mixture such as storage modulus (*E′* = 1260 MPa), glass transition temperature (*T*_g_ = 50.0 °C), or polymerization activation energy (17.3 kJ/mol) are enhanced in comparison with the methacrylated vegetable oil. The synthesized one-pot mixture does not require additional methacrylic acid due to the use of the one formed in the first step of the reaction, while the eventual thermoset mixture exhibits enhanced material properties compared to the methacrylated vegetable oil itself. Precursors synthesized in this work may find their purpose in the field of coating technologies, since these applications require detailed viscosity modifications.

## 1. Introduction

Various thermosets using bio-based templates have been synthesized recently, as the interest in using bio-based curable precursors (i.e., via thermal or UV-irradiation initiation) has been rising in recent decades [1,2,3,4]. The main reason several lipids or polyols are used for the production of polymerizable monomers is the applicability of their available naturally formed carbon structures [5,6,7]. The direct synthesis of numerous molecules, which can be obtained from agricultural productions would be complicated and, in most cases, economically unprofitable [5,8]. On the other hand, bio-source curable thermosets, such as acrylated or methacrylated vegetable oils (soybean, tung, linseed) [9,10], methacrylated polyols (pentaerythritol, isosorbide) [11], or lipid-based polyurethanes cannot just provide interesting carbon backbones to be modified but the reactants for their synthesis are often available and convenient to use [12]. There is a wide spectrum of usage for polymerizable precursors formed from renewable sources. Three-dimensional printing technologies, such as stereolithography (SLA), can incorporate these precursors into their products to increase their bio-content [13]. Several coating applications may benefit from the properties these materials can provide, namely paper coating or incorporation into controlled-released fertilizers (CRFs) [14,15].

Vegetable oils are used as templates due to the unsaturated double bonds (C=C) that they contain. These double bonds can be efficiently modified via epoxidation mixtures (typically H_2_O_2_ with carboxylic and catalyst acid) to epoxy functional groups (composed of bonded oxygen atoms in the oxirane cycle). Formed epoxidized oil is modified for example by polymerizable carboxylic acids (a type of nucleophile) such as acrylic or methacrylic acid. Typically, organic basic catalysts (e.g., aliphatic amines) are added to the solution to enforce the nucleophilic character of carboxylic acid [16,17,18]. Methyl lactate is a molecule possessing a hydroxyl functional group that can also be modified via an appropriate reactant. Carboxylic acid anhydrides, or acylhalogenides, are mostly used for this type of modification. The result of the methacrylation of hydroxyl groups by methacrylic anhydride is the particular ester of alcohol and methacrylic acid and there is also leftover acid formed as a secondary product of this reaction. This formed by-product is usually separated from the mixture due to its acidity and undesirable properties (dissociation, water solubility, odor, etc.) [19].

Generally, syntheses based on the usage of methacrylated anhydride use 4-dimethylaminopyridine (DMAP) as a catalyst [20,21,22,23]. However, this compound exhibits considerable toxicity [24] and it is also expensive. The present article describes the methacrylation process involving potassium 2-ethylhexanoate as a catalyst instead of the conventionally used DMAP-ethylhexanoic acid and potassium hydroxide, which are safer to handle during manufacturing processes [25] in comparison with DMAP [26]. Additionally, the catalytic properties of potassium 2-ethylhexanoate have not been investigated thoroughly in the field of methacrylic modification processes. Therefore, the experimental results regarding potassium 2-ethylhexanoate’s catalytic activity in connection with the reactions involving methacrylic anhydride could be considered resourceful.

This work aims to investigate the production of potential coating thermosetting precursors based on both methacrylated methyl lactates in a mixture with methacrylated rapeseed oil. The synthesis of methacrylated methyl lactate involving methacrylic anhydride as a reactant has not been widely investigated. A particular patent involving the esterification process using methacrylic acid has already been published [27]. The effectiveness and profitability of the presented production should be ensured using the formed by-product of methyl lactate’s methacrylation—methacrylic acid—and in the same batch, the methacrylation of the epoxidized rapeseed oil (prepared separately) will take place using the previously formed acid. The confirmation of successfully formed products shall be provided by FT-IR and NMR methods. An additional characterization of the produced precursor mixture will be performed including viscometry, differential scanning calorimetry (DSC), thermo-gravimetric analysis (TGA), or dynamic mechanical analysis (DMA) measurements.

## 2. Materials and Methods

### 2.1. Materials

Rapeseed oil for the preparation of synthesized oil along with hydrogen peroxide (H_2_O_2_; 35% *w*/*w*) was supplied by FICHEMA s.r.o. Czech Republic. Methyl (*R*)-(+)-lactate for the methacryled methyl lactate synthesis was obtained from Hefei Home Sunshine Pharmaceutical Technology Co., Ltd., China. All other reactants used for the mixture thermoset synthesis were acquired from Sigma Aldrich, particularly: Formic acid (puriss. p.a.), triethylamine (≥99%), 4-Methoxyphenol (for synthesis), methacrylic anhydride (94%); potassium hydroxide (p.a.), 2-ethylhexanoic acid (for synthesis), potassium iodide (ACS reagent, ≥99.0%), and sodium thiosulfate (ReagentPlus^®^, 99%). The solvent for NMR analyses (d-chloroform (CDCl_3_; 99.8%)) was obtained from Sigma Aldrich as well.

### 2.2. Synthesis of Curable Bio-Based Thermosets

The epoxidation of rapeseed oil was the first reaction performed. The oil characterization via volumetric analyses was performed (iodine value (I.V.), oxirane oxygen content (OOC), peroxide value (P.V), and acid value (A.V)) before any further modifications were made. Reactants were oil with measured I.V. (912 g) along with hydrogen peroxide (1.4 mol to I.V. of oil; 611.3 g) and formic acid as catalyst (0.25 mol to I.V. of oil; 51.5 g). The epoxidation reaction took place in a reactor of 2 L volume equipped with a shaft stirrer and a heating jacket. The oil was first heated to 43 °C. Then, the epoxidation mixture of H_2_O_2_ with HCOOH was added. Although the reaction was set to 62 °C, the epoxidation mixture was combined with the oil in advance due to the exothermic character of the reaction. The reaction was monitored for 10 h and the samples for continual I.V. and OOC analyses were collected (they were centrifuged to separate the water phase from the oil). The purification process was performed after the epoxidation. Modified oil was separated from the water phase and the reduction of free and bonded peroxides was realized via potassium iodide. The eventual separation of either the formed iodine molecule (from the reduction reaction) or the extra potassium iodide was performed by a water solution of sodium thiosulfate. The reaction scheme is illustrated in Figure 1.

Simultaneously, methyl lactate (1 mol, 104.1 g) was mixed with methacrylic anhydride (1 mol, 154.2 g) in a round bottom flask. The catalyst (50% *w*/*w* solution of potassium 2-ethylhexanoate in 2-ethylhexanoic acid (0.05 mol, 18.2 g of solution)) was added as soon as the mixture was tempered to 80°C. The reaction was performed for 10 h in an opened flask placed in a tempered oil bath. The conversion was quantified via GC method (Hewlett Packard 5890 Series II, Palo Alto, CA, USA; FID detector, column ZB-624, N_2_ as auxiliary gas for FIC; air (oxidizer for FID); H_2_ (carrier gas for FID); inlet temperature 200 °C; detector temperature 260 °C; temperature gradient with a rate of 20 °C/min for 28 min up to 250 °C); RT of methacrylated methyl lactate: 9.80 min). All reagents formed after the reaction remained in the flask. The methacrylation of methyl lactate is shown in Figure 2.

Methacrylated methyl lactate (MeLaMMA): ^1^H NMR (Appendix A) (CDCl_3_, 500 MHz): δ(ppm)6.18 (p; *J* = 1.08; 1.08; 1.07; 1.07 Hz; 1H), 5.62–5.61 (p; *J* = 1.63; 1.63; 1.61; 1.61 Hz; 1H), 5.15–5.11 (q; *J* = 7.05; 7.05; 7.05 Hz; 1H), 3.73 (s; 3H), 1.95 (t; *J*= 1.32; 1.32 Hz; 3H), 1.53(d; *J*= 7.08 Hz; 3H).

Methacrylic acid (MA):^1^H NMR (Appendix A) (CDCl_3_, 500 MHz): δ(ppm) 6.26–6.25 (dd; J = 1.52; 0.95 Hz; 1H), 5.69–5.67 (p; J = 1.66; 1.66; 1.63; 1.63 Hz; 1H), 1.96 (dd; J = 1.63; 1.01 Hz; 3H).

The formed methacrylic acid from the methyl lactate methacrylation was used as a nucleophile for the methacrylation of epoxidized rapeseed oil prepared earlier. Once the OOC parameter of the epoxidized rapeseed oil was determined, the respective molar amount of the oil was added to the mixture already containing both methacryled methyl lactate and methacrylic acid (and the methacrylation catalyst). The mixtures were prepared in three different molar ratios (epoxy groups: methacrylic acid) to observe the differences in reaction rates and conversions. The chosen ratios were 1:1 (253.0 g of epoxidized oil), 1:1.5 (189.8 g of epoxidized oil), and 1:2 (126.5 g of epoxidized oil), which means the presence of an equimolar amount or excess of methacrylic acid. The solution possessing all components was tempered at 95 °C and an additional polymerization inhibitor was added (4-methoxyphenol, 0.01 mol; 1.25 g). The reaction catalyst was poured into the mixture once the inhibitor was completely dissolved (triethylamine, 0.05 mol; 5 g). The nucleophilic substitution (methacrylation of epoxidized oil) was performed for 24 h. The eventual mixture containing formed methacrylated methyl lactate and methacrylated rapeseed oil was water-washed to remove all catalysts from the system and the remaining water was removed via dryer (sodium sulfate). The methacrylation of epoxidized rapeseed oil is illustrated in Figure 3. The whole preparation of the thermoset mixture is shown in Figure 1.

### 2.3. Structural Analyses of Synthesized Products

#### 2.3.1. Fourier Transformed Infrared Spectrometry (FT-IR)

Infrared spectrometry was used for the identification of processes that occurred during the reactions (modification of epoxy functional groups, the disappearance of hydroxyl functional groups, etc.). The used instrumentation was an infrared spectrometer Bruker Tensor 27 (Billerica, MA, USA), applied method was attenuated total reflectance (ATR) using diamond as a dispersion component. The diode laser served as an irradiation source. Michelson interferometer was used for the quantification of the signal. Spectra were composed out of 32 total scans with a measurement resolution of 4 cm^−1^.

#### 2.3.2. Nuclear Magnetic Resonance (NMR)

Nuclear magnetic resonance spectra were obtained by instrument Bruker Avance III 500 MHz (Bruker, Billerica, MA, USA) with the measuring frequency of 500 MHz for ^1^H NMR at the temperature of 30 °C using d-chloroform (CDCl_3_) as a solvent with tetramethylsilane (TMS) as an internal standard. The chemical shifts (*δ*) are shown in part per million (ppm) units. Coupling constant *J* has (Hz) unit with coupling expressed as s-singlet, d-doublet, t-triplet, q-quartet, p-quintet, m-multiplet.

#### 2.3.3. Volumetric Analyses

Acid value (A.V.) quantifies the amount of acidic functional groups. It is used for the verification of the non-acidic character of reactants and products after syntheses as well as for the quantification of methacrylation’s conversion. The sample (0.1–0.3 g) is diluted in the appropriate solution (acetone), pH indicator (bromothymol blue) is added to the mixture and 0.1 M potassium hydroxide in methanol is used as a titration solution. The calculation is shown in Equation (1):(1)$$A.V.=\frac{c_{KOH}\cdot V_{KOH}\cdot 56,100}{m_{sample}},$$
where A.V. is the acidic value (mg *KOH*/g), *c_KOH_* is the molar concentration of the titration solution (mol/dm^3^), *V_KOH_* is the volume of the titration solution (dm^3^) and *m_sample_* is the weight of the measured sample (g).

Peroxide value (P.V.) determines the occurring peroxides within the oil structure. The determination’s principle is to use potassium iodide as a reduction agent producing an iodine molecule as an oxidized product after the reduction of peroxides. This parameter is significant for the potential polymerizable system due to their possible spontaneous reactivity when peroxides are present. The determination requires an analyte sample (0.1–0.5 g) to be mixed with 5 mL of saturated potassium iodide solution. The solution is mixed for 1 min and is titrated with 0.01 M solution of sodium thiosulfate. Starch suspension can be used as an indicator. The peroxide value calculation is shown in Equation (2):(2)$$P.V.=\frac{c_{Na_{2}S_{2}O_{3}}\cdot V_{Na_{2}S_{2}O_{3}}}{m_{sample}}\cdot 100,000,$$
where P.V. is peroxide value (μmol O_2_/g), *c_Na2S2O3_* is the molar concentration of the titration solution (mol/dm^3^), *V_Na2S2O3_* is the volume of the titration solution (dm^3^) and *m_sample_* is the weight of the measured sample (g).

Iodine value (I.V.) is a parameter for the determination of the number of unsaturated bonds within oil structures. Various reactants can be used for the determination of iodine monochloride (ICl). The procedure involves sample weighing (0.1–0.3 g) and mixing it with 20 mL of 0.1 M ICl solution in acetic acid. The prepared mixture requires 30 min for the reaction. The solution is mixed with 20 mL of 10% *w*/*w* solution of potassium iodide in water to force the quantitative formation of iodine molecules. The system is titrated with 0.1 M solution of sodium thiosulfate. A starch solution can be used as an indicator. The calculation of iodine value is illustrated in Equation (3):(3)$$I.V.=\frac{\left(V_{BLANK}-V_{Na_{2}S_{2}O_{3}}\right)\cdot c_{Na_{2}S_{2}O_{3}}\cdot 126.9\cdot 100}{m_{sample}},$$
where I.V. is iodine value (g I_2_/100 g), *c_Na2S2O3_* is the molar concentration of the titration solution (mol/dm^3^), *V_BLANK_* is the volume of the titration solution for blank (dm^3^), *V_Na2S2O3_* is the volume of titration the solution for sample (dm^3^), and *m_sample_* is the weight of the measured sample (g).

Hydroxyl value (H.V.) is a quantity of hydroxyl functional groups occurring in a chemical structure. The principle of determination is the acetylation of vacant hydroxyl groups via acetylic anhydride in the presence of pyridine as a catalyst. The sample (0.25–0.5 g) is mixed with 5 mL of 25% *w*/*w* solution of acetylic anhydride in pyridine. The mixture is tempered at 100 °C for 1 h. The solution is mixed with 10 mL of water once the reaction is performed to hydrolyze excessing anhydride. The mixture is titrated with 1 M potassium hydroxide solution in water. Bromothymol blue is used as an indicator. The calculation of hydroxyl value is provided in Equation (4):(4)$$H.V.=\frac{\left(V_{BLANK}-V_{KOH}\right)\cdot c_{KOH}\cdot 56,100}{m_{sample}},$$
where H.V. is hydroxyl value (mg KOH/g), *c_KOH_* is the molar concentration of the titration solution (mol/dm^3^), *V_BLANK_* is the volume of the titration solution for blank (dm^3^), *V_KOH_* is the volume of titration the solution for sample (dm^3^) and *m_sample_* is the weight of the measured sample (g).

Oxirane oxygen content (OOC) quantifies the percentage amount of cyclic bonded oxygen as a result of the oxidation of double bonds. The determination of this parameter is provided via the nucleophilic substitution reaction of hydrobromic acid (HBr) and epoxy functional groups. The proton releases the cyclic structure of the epoxy group and nucleophile (Br^−^) attacks formed carbocation. Analysis proceeds as follows. The sample (0.1–0.3 g) is added to the titration flask and 10 mL of 99% *w*/*w* acetic acid is added. This solution is enriched with 5 drops of crystal violet water solution (indicator) and titrated via 0.1 M solution of HBr in acetic acid. The calculation of oxirane oxygen content is written in Equation (5):(5)$$\mathrm{OOC}=\frac{V_{HBr}\cdot c_{HBr}\cdot 1.6}{m_{sample}},$$
where OOC is oxirane oxygen content (%), *c_HBr_* is the molar concentration of the titration solution (mol/dm^3^), *V_HBr_* is the volume of titration of the solution for sample (cm^3^) and *m_sample_* is the weight of the measured sample (g).

### 2.4. Characterization of Physical-Chemical Properties

#### 2.4.1. Viscometry

Apparent viscosity (η_app_) is an important parameter for the description of the system’s behavior prediction, especially at the processing temperature range. The viscosity measurements were completed via digital viscometer DV-II+ PRO EXTRA (BROOKFIELD, Middleboro, MA, USA). The determination of apparent viscosity required 50–80 mL samples and various rheology shafts were used depending on the viscosity level of each measured system.

#### 2.4.2. Differential Scanning Calorimetry (DSC)

Differential scanning calorimetry (DSC) served as a confirming method of the curability of synthesized precursors. The synthesized molecules were mixed with Luperox^®^ DI, *tert*-Butyl peroxide as thermo-initiator (in 1% *w*/*w* quantity to precursor). The samples consisted of 10 g of each precursor and 0.1 g of initiator which was added to the mixture and dissolved. The mixtures were placed in aluminum pans and hermetically sealed. DSC 2500 model from TA instruments (New Castle, DE, USA) was used for the analyses. Samples were subjected to a heating scan from 10 to 240 °C with differing temperature ramps: 5, 10, 15, and 20 °C/min.

#### 2.4.3. Dynamic Mechanical Analysis (DMA)

The viscoelastic properties of prepared materials were measured using a dynamical mechanical analyzer, DMA 2980 from TA Instruments (New Castle, DE, USA). Testing specimens with typical dimensions of 60 × 10 × 2 mm were prepared by casting the resin with a photoinitiator into the mold and curing. Polymeric samples were prepared with methacryled rapeseed oil (MRO), methyl lactate methacrylate (MeLaMMA), and their mixture was prepared in a one-pot synthesis. Phenylbis(2,4,6-trimethylbenzoyl)phosphine oxide (BAPO) as a photoinitiator was added to each precursor in 1% *w*/*w*. The samples consisted of 20 g of each precursor and 0.2 g of initiator, which was added to the mixture and dissolved. The irradiation source with a wavelength of 405 nm was used for the initiation for 30 min. The specimens were mounted into dual cantilever geometry and subjected to a cyclic deformation with 10 μm and 1 Hz. The samples were heated from laboratory temperature to 120 °C with a heating rate of 3 °C/min.

#### 2.4.4. Thermo-Gravimetric Analysis (TGA)

Thermo-gravimetric analysis (TGA) was used for the determination of the heat stability index of prepared precursors which were polymerized. Polymeric samples were prepared the same way as samples for DMA analysis. TGA analysis itself was performed on TGA Q500 from TA Instruments (New Castle, DE, USA). The degradation process of a sample (10–15 mg) was monitored via the following heating conditions: equilibration at 40 °C; heating to 600 °C at a heating rate of 10 °C/min under N_2_; 10 min at 600 °C under air atmosphere.

## 3. Results

### 3.1. Polymerizable Thermoset Mixture Synthesis

The analysis of used rapeseed oil was performed before it was epoxidized due to the fact that the information on iodine value is especially necessary for the preparation of the reaction mixture. Reactants that undergo the epoxidation reaction (hydrogen peroxide and formic acid) are calculated according to the iodine value as a parameter determining the amount of modifiable double bonds. The results of all measured volumetric parameters are shown in Table 1.

The used rapeseed oil had a particular composition of fatty acid within its structure. This factor might be important for comparison with other vegetable oils modified the same way. However, only the oil with the highest ratio of mono-unsaturated fatty acids has been studied in this work. The particular composition of occurring fatty acids is displayed in Table 2. The saturated fatty acid share is an important parameter for the used vegetable oil since this type of structure cannot be modified for further polymerizable purposes.

The epoxidation mixture prepared from reactants according to the determined iodine value of rapeseed oil underwent a reaction for 10 h. Both iodine value and OOC parameters were monitored during the reaction for the confirmation of forming epoxy functional groups within the structure of the used oil. The OOC parameter increases over time, as is shown in Figure 2a, as a result of forming of oxirane cycles. On the other hand, double bonds disappear from the carbon backbone of used oil as an iodine value decreases. The mixture was treated with potassium iodide and sodium thiosulfate after the reaction to reduce the number of peroxide functional groups occurring in the oil’s structure after the epoxidation. The solution was eventually washed with distilled water to remove all formed structures and the remaining hydrogen peroxide and formic acid. Finally, the epoxidized oil was distilled to separate residual water from the emulsion.

There is evidence of positive progress of the epoxidation from Figure 2a. The OOC parameter raised to the value of 6.23%. Since the maximal OOC which can be reached with the initial iodine value of 120.5 g I_2_/100 g is 7.22%, the conversion of double bonds into epoxy groups is 86.3% according to OOC. The iodine value decreased in time during the reaction and reached a value of 17.6 g I_2_/100 g. If the conversion of double bonds is calculated via the iodine value, the result is 86.5%. The value calculated from the decrease in iodine value is slightly higher than the one from OOC. The reason for this outcome is the fact that a part of the double bonds participates in the formation of peroxide functional groups within the oil’s structure. However, both parameters correlate. It was observed that when the oil undergoes the reaction for a longer time, the OOC parameter tends to decrease. Therefore, the time of 10 h was determined as optimal. Published epoxidation processes involving oils such as camelina oil, flax oil, or canola oil reached similar or lower OOC parameter values in comparison with presented results, except for the reaction time, which was longer [28]. It was found that 10 h is an optimal reaction time, particularly for rapeseed oil.

The epoxidation process appeared to be exothermic after the addition of the catalyst. The temperature increase is an important factor for the potential scale-up and production technology suggestions. The heating jacket of the reactor can temper the temperature and cool down the system. However, the amount of released energy during this process is non-negligible. The temperature increase after the catalyst’s addition and the following cool-down is shown in Appendix A.

Separately, the synthesis of methacrylated methyl lactate (MeLaMMA) was performed to generate both polymerizable monomer (MeLaMMA) and reaction methacrylic acid (MA) for a further methacrylated oil synthesis (illustrated in Figure 1). The polymerizable monomer forms a continuum for an additional reaction step as well as methacrylic acid. The reaction’s progress in time is shown in Figure 2, which is a progress of the MAA concentration’s decrease and MA concentration’s increase in time monitored via the GC-FID method.

There is evidence of a reaction’s progressing trend from the chromatography results in Figure 2b. Methacrylic anhydride reacts with methyl lactate (MeLa), so its concentration in the reaction mixture decreases over time. At the same time, methacrylic acid is formed as one of the products during the reaction, which illustrates the dependence above as well. Since the methacrylation reaction of MeLa and MAA does not require the withdrawing of the forming by-product MA to modify the equilibrium, this reaction was performed until the maximum conversion of both reactants (MA and MeLaMMA) was acquired. The reached yield values of both products are shown in Table 3.

The post-reaction mixture containing both formed MeLaMMA and MA was heated from 80 °C to 95 °C before the addition of prepared epoxidized oil and appropriate catalyst (TEA). An additional inhibitor (4-methoxyphenol) was dissolved in the solution to prevent the mixture from spontaneous polymerization. This additive is also present in used methacrylic anhydride at 2000 ppm. However, since the thermoset mixture preparation process took two steps both involving increased temperature conditions, the amount of inhibitor increased. All three chosen molar ratios of reactants were monitored during the 24 h long reaction and the decreases of OOC parameters were measured for all of them regularly. The molar ratios contain an either equimolar amount of MA (1:1 mol) or an excess of MA (1:1.5 mol; 1:2 mol). The dependences of OOC parameter changes in time and calculated epoxy functional groups conversions from the OOC parameters’ theoretical and real values are shown in Figure 3a,b.

A big influence on the methacrylation reaction rate (of epoxidized oil) is caused by particular excess of methacrylic acid. Figure 3a shows the OOC parameter values decrease in time for all performed mixtures. The initial OOC parameter differs in the graph due to the unequal mass participation of oil in studied mixtures (the equimolar mixture has twice the amount of epoxidized oil than the solution with double MA excess). Therefore, the results comparing OOC values do not show an accurate reaction conversion. On the other hand, Figure 3b is composed of the particular dependences of epoxy functional groups conversions in time during the reaction. The conversion values were calculated as follows:(6)$$\mathrm{Epoxy}\;\mathrm{groups}\;\mathrm{conversion}=\left(1-\frac{{\mathrm{OOC}}_{t}}{{\mathrm{OOC}}_{0}}\right)\times 100,$$
where OOC_t_ is the value of epoxy groups percentage in particular time t (%), OOC_0_ is the value of epoxy groups percentage in time t= 0 min (%). The conversions reached after 24 h of reaction for each mixture containing different amounts of MA excess are shown in Table 4.

The resulting mixture containing both methacrylated rapeseed oil and methacrylated methyl lactate exhibited particular oil numbers as a confirmation of performed modifications and reactions. The values of measured parameters are written in Table 5. There is an evident presence of hydroxyl functional groups from the hydroxyl value confirming the bonded methacrylic functional groups in the oil’s structure. This bonding is also evident from the raised iodine value. The remaining OOC parameter value is a measurement of the reaction conversion and gives information about possible further modification performance possibilities. However, the mixture was set at 95 °C to modify the epoxidized oil for only 24 h so the solution does not spontaneously polymerize. When the calculated conversions are compared to the reported results using epodixized soybean oil and acrylic acid the excess of (1:10) (oil:acid), the final products reached a similar conversion as the reported ones. Therefore, the used excesses of the acid investigated in the work should increase the profitability of the process [29].

### 3.2. Structural Characterization

#### 3.2.1. Fourier Transformed Infrared Spectrometry (FT-IR)

Infrared spectrometry served as proof of specific functional groups within the structures. In particular, the formation of hydroxyl groups can be better confirmed via the FT-IR method than the NMR (measured in CDCl_3_) due to the fact the nuclear magnetic resonance in CDCl_3_ does not show hydroxyl groups reliably. All infrared spectra of reactants and products of the one-pot synthesis of the thermoset mixture are shown in Figure 4.

The disappearance of signals of unsaturated bonds within the rapeseed oil (RO) structure (peak b) is evident. There is a signal belonging to the epoxy functional groups (peak c) in the structure of epoxidized rapeseed oil (ERO). The final product containing both methacrylated rapeseed oil (MRO) and methacrylated methyl lactate (MeLaMMA) shows both signals for hydroxyl groups (peak e) and regained signals for unsaturated bonds which belong to methacrylate groups (peak d).

#### 3.2.2. Nuclear Magnetic Resonance

Nuclear magnetic resonance (NMR) was used for the hydrogen-containing carbon structure confirmation of synthesized products and used reactants. The method was used mainly for the confirmation of the appearance and disappearance of particular signals which belong to hydrogen atoms in certain structural positions. The NMR spectra were not integrated and used for the calculation of an exact number of hydrogen atoms since the oil structure is not exactly defined and contains multiple different fatty acid structures. However, the particular signals which refer to certain functional groups (provided via hydrogen atom signal position) can be found in spectra. All spectra of reactants and products of the one-pot synthesis of the thermoset mixture are shown in Figure 5.

The most evident confirmation of the epoxidation of rapeseed oil (RO) is the appearance of the signal referring to epoxy functional groups (peaks g) as well as the disappearance of unsaturated bonds signals (peaks a and e) in the structure of epoxidized rapeseed oil (ERO). The further modification of oil resulting in the synthesis of methacrylated rapeseed oil (MRO) and methacrylated methyl lactate (MeLaMMA), as a previously formed molecule, is proved by the appearance of other peaks described in the spectrum (c).

### 3.3. Physical-Chemical Properties Characterization

#### 3.3.1. Viscometry

The monitoring of the rheological properties of synthesized precursors is essential for their complete description and suggestion for further usage. The viscosity of polymerizable precursor is one of the most important factors due to, e.g., its enormous impact on the thickness of formed coated layers. The synthesis and production process itself relies on the viscosities of particular components in the reaction. The dependences of apparent viscosities of each reactant and product of the whole one-pot multiple steps synthesis on increasing temperature were investigated to closely describe their behavior during the working process or their storage. The dependencies are shown in Figure 6. It is evident that methacrylated rapeseed oil (MRO) has the highest values of apparent viscosity in comparison with other molecules. It is the consequence of the presence of hydroxyl functional groups in its structure (as FT-TR, NMR, and H.V. confirm as well). This functional group is capable of forming hydrogen bonding, therefore more intermolecular interactions are formed, which increases the cohesion of the precursor and its viscosity. The epoxidized rapeseed oil (ERO) has lower values of viscosity than MRO but higher than crude rapeseed oil (RO) and the mixture of MRO and methyl lactate methacrylate (MeLaMMA) as a product of one-pot synthesis using double excess of methacrylic acid (ratio of epoxy groups to MA was 1:2). The epoxy functional groups cannot form hydrogen bonding but their ability to exhibit more Keesom and Debye interactions (relying on permanent dipole moment of a molecule) causes their viscosity’s increase. Reported viscosities of epoxidized and acrylated soybean oil, which are similar to rapeseed oil from the structural point, reach the same viscosities as the presented synthesized materials [30].

The dependences of apparent viscosity on increasing temperature obeying the Arrhenius equation were investigated. We can find a basic exponential model describing the decrease of apparent viscosity with increasing temperature [31]. According to the literature, the effect on viscosity by increasing temperature is described by Equation (7):(7)$$\eta =\eta_{\infty }e^{\frac{E_{a}}{RT}},$$
where *η* stands for apparent viscosity (in general, for Newtonian fluids it stands for dynamic viscosity) (Pa·s); *η*_∞_ is a pre-exponential factor which is usually considered to be an infinite-temperature viscosity (Pa·s); *E_a_* is exponential constant and often described as activation energy (J/mol); R is the gas constant (J/(mol·K)) and *T* is the thermodynamic temperature (K). Usually, this equation is rearranged using a logarithm addition to generate the linear function which describes the following equation:(8)$$\ln\eta =\frac{E_{a}}{R}\cdot \frac{1}{T}+\ln\eta_{\infty },$$
when the dependence of ln (*η*) on 1/*T* is constructed, we can obtain the activation energy (*E_a_*) from the slope by multiplying it by R, and also, we can extract the infinite-temperature viscosity (*η*_∞_) from the *y*-intercept by applying exponential operation.

The Arrhenius graphs are displayed in Appendix A, and the calculation of particular quantities is written in Table 6. The one-pot mixture described in the following graph and table is consisted of methacrylated rapeseed oil (MRO) and methyl lactate methacrylate (MeLaMMA) after the reaction using double molar methacrylic acid excess (1:2 mol) (epoxy groups:MA).

MeLaMMA decreases the apparent viscosity of the mixture with MRO tremendously. The mass content of MeLaMMA in a mixture is approximately 45% *w*/*w* after the one-pot reaction. Additionally, the dependence of the apparent viscosity of the system on the mass content of MeLaMMA in the mixture was investigated since the methacrylated methyl lactate could be synthesized separately and worked as a viscosity modifier. The measured dependence is shown in Figure 7. The dependence was measured and constructed from zero concentration of MeLaMMA up to the amount resulting in the one-pot synthesis process (45% *w*/*w* content of MeLaMMA), because higher excess of MA during the synthesis process is not needed.

#### 3.3.2. Polymerization Activity

Synthesized precursors were analyzed via differential scanning calorimetry (DSC) to obtain data regarding their polymerization reactivity. The main aim was to determine whether the produced one-pot thermoset mixture exhibits a better or worse ability to be polymerized. The investigation consisted of mixing the precursor with a thermal initiator and then measuring all mixtures using different temperature-increasing rates to calculate parameters occurring in the equation referring to Kissinger’s theory. According to Kissinger’s theory, the heating rate (*β*) and measured values regarding exothermic peak temperatures (*T_p_*) are present in the following equation [32]:(9)$$\ln\left(\frac{\beta }{T_{p}{}^{2}}\right)=\ln\left(\frac{AR}{E}\right)-\frac{E}{R}\cdot \frac{1}{T_{p}},$$
where *β* is the heating rate (°C/min); *T_p_* is exothermic peak temperature (°C); *A* is a pre-exponential factor (-); *E* is the activation energy of the reaction (J/mol) and *R* is the gas constant (J/(mol·K)). The results of calorimetric analyses are graphically illustrated in Figure 8.

From the thermograms, it is evident that methacrylated rapeseed oil (MRO) exhibits the lowest reactivity considering a thermal-initiation polymerization. The peak temperature of MRO reaches a higher value (179.1 °C) than MeLaMMA (157.3 °C). The one-pot synthesized resin mixture containing both polymerizable monomers (MRO and MeLaMMA) exhibited the lowest peak temperature at 151.0 °C during the DSC analysis at a 10 °C/min heating rate. The shown DSC curves confirm that mixing MRO with MeLaMMA during their synthesis has a beneficial effect on the mixture’s polymerization reactivity. A higher reactivity connected to the fast enthalpy release of MeLaMMA is caused by the mono-functional polymerizable structure of MeLaMMA. In the thermogram of the mixture, the initiation process reached a faster rate for sterically available structures than for poly-functional MRO composed of the complex carbon backbone, which is manifested as a peak tail around 180 °C (which corresponds to pure MRO). The measured values of the peak temperatures for each produced precursor at different heating rates and the calculated parameters (activation energy *E* and pre-exponential factor *A*) are shown in Table 7 and graphically illustrated in Appendix A.

#### 3.3.3. Dynamic Mechanical Properties

The dynamic mechanical properties of cures thermoset consisting of synthesized polymerized precursors were measured. The glass transition temperature (*T*_g_), which is connected to the loss of molecules’ solid phase structure organization resulting in the sudden decrease of the material’s rigidity, was taken as a maximum of loss factor (tan δ). The cross-linking density (*ν_e_*, mol/m^3^) is a thermoset property calculated from the rubbery elastic theory which includes following equation [33]:(10)$$\upsilon_{e}=\frac{E^{\prime }}{3RT^{\prime }},$$
where *E′* stands for the storage modulus in the rubbery plateau region (E′ at *T*_g_+ 40 °C) (Pa); *R* is the gas constant (J/(mol·K)) and *T′* is the thermodynamic temperature (at *T*_g_ + 40 °C) (K). The dependence of the loss factor on temperature together with the Wicket plot of the samples is shown in Figure 9 and the calculated and determined parameters defined above (storage modulus, glass transition temperature, and cross-linking density) are shown in Table 8. The graphs containing dependences of the storage modulus and the loss modulus on temperature regarding measured thermosets are shown in the Appendix A (Appendix A). As can be seen from the Wicket plot of the samples, the data follow a general relationship between log (loss factor) and log(modulus) of inverted “U” shape with the exception of a few marginal points of the samples MRO and MeLaMMa + MRO. The data are reliable and obey the WLF assumption.

According to the results, MRO is supposed to have the highest cross-linking density (56.5 kmol/m^3^) of all produced thermosets. However, the storage modulus of MRO reached the lowest value (640 MPa), which is due to the fact that compared storage modulus was taken at 30 °C. At this temperature, MRO is above its *T*_g_ reaching a rubbery plateau of storage modulus. On the other hand, MeLaMMA possesses the lowest cross-linking density (37.4 kmol/m^3^) and the highest storage modulus (2610 MPa). The mixture containing approximately 45% *w*/*w* content of MeLaMMA mixed with MRO exhibited values of the cross-linking density and storage modulus in between both individual components (*ν_e_*= 44.9 kmol/m^3^; E′ = 1260 MPa). The highest glass transition temperature (*T*_g_) was measured for MeLaMMA (68.8 °C) and in the mixture with MRO the *T*_g_ of the system decreased (having an average of 50.0 °C). The highest value of cross-linking density for MRO might be the result of the high functionality of the methacrylated oil structure (on average, the structure contains approximately three polymerizable methacrylic functional groups). On the other hand, the functionality of MeLaMMA is equal to one and the system exhibited a cross-linked structure due to the BAPO initiator present during its curing. The decrease of a cross-linking density can enhance the storage modulus, since the values of *E′* for the one-pot mixture increased and so did the material properties of MeLaMMA alone. The higher storage modulus of MeLaMMA in the combination with the lower cross-linking density could be the result of the structural characteristics of MeLaMMA. Since the storage modulus of the material represents the elastic component of the system, MeLaMMA could be forming a more cured polymeric structure due to its high reactivity and mainly due to its smaller size. The DMA analysis confirmed that not only can MeLaMMA serve as a viscosity modifier but its presence in the mixture enhances the dynamic mechanical properties of the cured thermoset. The results are promising for industrially scalable applications, since the process uses the majority of methacrylic anhydride mass potential through the process, while the viscosity can be directly modified, and the material properties are enhanced at the same time. Methacryled methyl lactate has been synthesized via esterification [32] which is a more difficult process from the energetic consumption standpoint than the one described in this work. Another reference describes the synthesis of MeLaMMA using methyl 2-chloropropionate while reaching the yield of 49% of MeLaMMA [35].

#### 3.3.4. Thermal Stability

Thermo-gravimetric analysis (TGA) was performed on cured thermosets in order to obtain data regarding their thermal stability. The results of the analyses were used for the calculation of the heat-resistant index (*T_s_*). The formula leading to the calculation of this parameter is written in Equation (11) [36]:(11)$$T_{s}=0.49\left[T_{5}+0.6\left(T_{30}-T_{5}\right)\right],$$
where *T_s_* stands for heat-resistant index (-); *T*_5_ is temperature at 5% of mass loss (°C) and *T*_30_ is temperature at 30% of mass loss (°C). The results and values involve both the calculated heat-resistant index and inflection point of mass loss (*T_max_*) at which the thermal degradation reaches its maximum. The dependence of the weight loss of the synthesized thermoset on increasing temperature and its derivation are shown in Figure 10. The calculated and measured data are summarized in Table 9.

The results confirm the outcome of DMA which proved that the methacrylated rapeseed oil (MRO) possesses the highest cross-linking density. According to the TGA measured values, the MRO reaches the highest heat-resistant index (*T*_s_) (*T*_s_ = 170.3) as well as the highest thermal stability expressed as the temperatures of 5% and 30% of mass loss. On the other hand, methacrylated methyl lactate (MeLaMMA) exhibits the lowest heat-resistant index (*T*_s_ = 130.9) and both the 5% and 30% mass loss temperatures. The weight loss of the MeLaMMA sample is undulating, reflecting the inhomogeneous structure of this material. This is further highlighted in the derivative curve, which contains several peaks where the temperature is not characteristic, and it changes when more samples are measured. The synthesized one-pot thermoset mixture reached a heat-resistant index in between both of its components (*T*_s_ = 146.0). Therefore, it has been confirmed that both the cross-linking density and heat-resistant index increase in the mixture of MRO and MeLaMMA in comparison with MeLaMMA.

## 4. Discussion

This work aimed to synthesize the thermoset mixture based on rapeseed oil, which was modified via epoxidation and then methacrylated using methacrylic acid formed as a secondary product of the methacrylation of methyl lactate using methacrylic anhydride. The epoxidation modification of used oil reached a conversion of 86.3% calculated from the measured OOC parameter of the product. Simultaneously, methacrylated methyl lactate was produced (98.25% of yield) along with the co-formed methacrylic acid (98.08% of yield). The epoxidized rapeseed oil was then added to the post-reaction mixture containing formed methacrylic acid to study the methacrylation process of an epoxy functional group within the oil’s structure. Three different reactants’ ratios were calculated to study the differences in the rates of the reaction processes. The solution containing double excess of methacrylic acid to epoxy groups exhibited the highest conversion of epoxy groups to methacrylate groups after 24 h (83.33%). On the other hand, the lowest conversion (23.08%) was reached with the mixture with an equimolar amount of both epoxy functional groups and methacrylic acid. This outcome confirms that the methacrylic excess in the mixture is significantly important to the reaction rate. All synthesized products were structurally analyzed via FT-IR and ^1^H NMR methods which verified all changes in the modified functional groups.

The viscometry of produced thermoset mixture containing both methacrylated rapeseed oil and methacrylated methyl lactate proved that the presence of modified methyl lactate decreases the apparent viscosity of synthesized thermoset tremendously. The apparent viscosity of 100% modified oil reached a value of 1790.2 mPa·s and the one-pot reaction mixture (containing approximately 45% *w*/*w* of methacrylated methyl lactate) exhibited a value of 142.6 mPa·s (both measured at 30 °C). Additionally, the polymerization kinetics of produced thermosets was investigated. The lowest activation energy calculated from Kissinger’s theory was reached by methacrylated methyl lactate (13.7 kJ/mol) and the highest by methacrylated rapeseed oil (20.7 kJ/mol). The one-pot synthesized mixture exhibited a decreased value of activation energy in comparison with the modified oil (17.3 kJ/mol). The dynamic mechanical analysis confirmed the increases of both storage modulus (1260 MPa) and glass transition temperature (50.0 °C) of the one-pot synthesized mixture in comparison with methacrylated rapeseed oil’s storage modulus (640 MPa) and glass transition temperature below the laboratory temperature (14–35 °C, according to literature). Lastly, the thermo-gravimetric analysis measurements uncovered that the lowest heat-resistant index (*T*_s_ = 130.9) was reached for methacrylated methyl lactate, while the methacrylated oil exhibited the highest (*T*_s_ = 170.3) and the one-pot mixture heat-resistant index was in between (*T*_s_ = 146.0) which means that the temperature resistance of synthesized mixture decreased in comparison with modified oil.

The potential for material application of prepared thermosetting mixture in the fields of coating technologies is promising based on the acquired results. The main aim of this work was to synthesize an oil-based polymerizable precursor whose viscosity can be directly modified. Simultaneously, the mixture consists of methacrylated methyl lactate, which is a novel compound in the field of material chemistry. The mechanical analyses performed in this work also confirmed that some properties, such as the storage modulus of the glass transition temperature, are improved using methacrylated methyl lactate in the system. A potentially profitable and scalable precursor mixture was synthesized and described and could be used in several material fields.

## 5. Conclusions

Several statements can be formulated according to the results obtained. The thermoset mixture containing both methacrylated methyl lactate and methacrylated vegetable oil can be synthesized in one batch involving a two steps process. In the first step, the methacrylated methyl lactate and methacrylic acid are produced following the second step in which epoxidized vegetable oil is added using the formed methacrylic acid for its modification. The produced one-pot mixture exhibits lower apparent viscosity than the methacrylated oil alone, which is an advantageous property for particular applications (e.g., coating technologies). The presence of methacrylated methyl lactate also enhances other properties of the system such as glass transition temperature, the storage modulus, and polymerization reactivity. The advantages of the presence of oil structure in the mixture are mainly a higher bio-based content, a higher cross-linking density, and an increased heat-resistance index in comparison with methyl lactate methacrylate.

## Data Availability

Not applicable.

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
