# Peer review of "Synthesis of Bio-Based Thermoset Mixture Composed of Methacrylated Rapeseed Oil and Methacrylated Methyl Lactate: One-Pot Synthesis Using Formed Methacrylic Acid as a Continual Reactant"

_polymers, 2023, doi:10.3390/polym15081811_

Round 1
Reviewer 1 Report
The present paper deals with synthesizing acrylate monomers based on rapeseed oil and methyl lactate.
It is well-discussed, while several issues exist.
These issues need to be clarified before the publishing of the present paper.
1) The novelty of the proposed synthesis approach is not stressed in the text. It looks like a routine process.
2) The details about the sample preparation are missing in Methods.
3) The text in the section Discussion is more like a reporting of data than a critical analysis. The data need to be discussed and referenced to the literature data.
4) Figures 2-4 can be combined in one image or even put in Supplementary. The number of figures needs to be decreased.
5) The viscosity in Figs 8 -9 needs to be in the log scale.
6) Please discuss why the crosslinking degree of the MeLaMMA is the lowest, while the storage modulus has the highest value.
7) section 4 is very vague.
Reviewer 2 Report
This papers describes a well designed sequence to obtain partially biobased methacrylate based thermosets based on mixtures of methacrylated rapeseed oil and methacrylated methyl lactate. The polymerisation mixture is obtained in a cleverly designed procedure which has clear potential for industrial application. The experiments have been carefully designed, described and interpreted. I recommend publication essentially as it stands.
Reviewer 3 Report
Abstract: The abstract should start with the research objectives, followed by the methodology, then main results and conclude with the possible application.
Keywords: Choose keywords other than main words in the title. It will improve the visibility of the published article.
Introduction: The introduction is very short. It should provide sufficient background about the current research and include all relevant recent references.
Results and Discussion:
What is the novelty of the work?
Results are not interpreted and discussed with sufficient literature.
The authors should clearly explain and compare the scientific reason for the obtained properties with existing recent reports.
DMA: Should have Storage and Loss modulus, Tan Delta. The homogeneity should be explained based on the Wicket plot.
TGA: Derivative thermograms must be provided and explained.
Conclusion: A conclusion must answer your problem statements as well as supported by the data gathered, followed by a concise take home message.
Round 2
Reviewer 1 Report
The paper was corrected according to the requested. It can be accepted.
Reviewer 3 Report
Authors have addressed all the comments satisfactorily. The manuscript may be accepted for publication in its present form.